# Biological and Cytoprotective Effect of *Piper kadsura* Ohwi against Hydrogen-Peroxide-Induced Oxidative Stress in Human SW1353 Cells

**DOI:** 10.3390/molecules26206287

**Published:** 2021-10-18

**Authors:** Te-Yang Huang, Chih-Chuan Wu, Wen-Ta Su

**Affiliations:** 1Department of Orthopedic Surgery, Mackay Memorial Hospital, Taipei 104217, Taiwan; haungt33@gmail.com; 2Department of Chemical Engineering and Biotechnology, National Taipei University of Technology, Taipei 104217, Taiwan; cf1811@ntut.edu.tw

**Keywords:** *Piper kadsura* Ohwi, microwave-assisted extraction, oxidative stress, SW1353, MAPKs pathways

## Abstract

Oxidative stress plays a role in regulating a variety of physiological functions in living organisms and in the pathogenesis of articular cartilage diseases. *Piper kadsura* Ohwi is a traditional Chinese medicine that is used as a treatment for rheumatic pain, and the extracts have anti-inflammatory and antioxidant effects. However, there is still no study related to cell protection by *P. kadsura*. The *P. kadsura* extracts (PKE) were obtained by microwave-assisted extraction, liquid-liquid extraction, and column chromatography separation. The extracts could effectively scavenge free radicals in the antioxidant test, the EC_50_ of extracts is approximately the same as vitamin C. PKE decreased the apoptosis of SW1353 cells treated with H_2_O_2_ and could upregulate the gene expression of antioxidant enzymes (SOD-2, GPx, and CAT) and the Bcl-2/Bax ratio, as well as regulate PARP, thus conferring resistance to H_2_O_2_ attack. PKE protects cells against apoptosis caused by free radicals through the three pathways of JNK, MEK/ERK, and p38 by treatment with MAPK inhibitor. The identified components of PKE were bicyclo [2.2.1] heptan-2-ol-1,7,7-trimethyl-,(1S-endo)-, alpha-humulene, and hydroxychavicol by gas chromatography–mass spectrometry.

## 1. Introduction

Osteoarthritis (OA) is a progressive degenerative joint disease that affects cartilage, synovium, synovial joints, and muscles around the joints [1,2]. OA is most affected by age, obesity, local inflammation, and mechanical stresses. The main pathological changes of osteoarthritis occur in articular cartilage, which also affects subchondral bone, tendon, joint periosteum, and surrounding muscle. After degenerative changes in cartilage, the appearance of the cartilage surface is irregular, with cracks, ulcers, cartilage thickness reduction, and joints. Inflammation of the synovial membrane results in asymmetrical stenosis of the joint cavity, joint pain, stiffness, and dysfunction. Clinically, it is caused by *extracellular matrix* (ECM) degradation and dysregulation of some inflammatory factors [3]. Unbalanced reactive oxygen species (ROS) may also destroy the structure of chondrocytes, enhance cell death signals, and reduce cell integrity, leading to cartilage damage. The dysregulation of ECM anabolism, abnormal generation of ROS, as well as overproduction of proteolytic enzymes and inflammatory cytokines can accelerate the degradation process of cartilage [4]. ROS are the byproducts of normal cellular metabolism and are thought to be essential for regulating physiological functions involved in the development and modulation of the immune system through activation of various cellular signaling pathways. A dramatic increase in ROS is called oxidative stress, which is an imbalance between the ratio of free radicals to antioxidants, which interferes with the normal redox state of the cell and damages the cell’s proteins, lipids, and DNA [5].

H_2_O_2_ is the major source of endogenous ROS; it has been extensively used to induce oxidative stress in in vitro models and to induce apoptosis in several types of cells, including chondrocyte and neuronal cells [6]. When ROS exceeds the ability of cells to load, the internal antioxidant capacity and oxidative capacity of the cells are unbalanced. This is called oxidative stress, and this result can destroy the normal structure of the mitochondria in the cell and its function, leading to death or apoptosis [7,8].

Studies on primary human chondrocytes are limited by the lack of a sufficient number from the donors. In addition, the isolated cells, primary human chondrocytes, very easily lose their biological and functional characterization in vitro culture and differ considerably with donors [9]. The human chondrosarcoma-derived cell line SW1353 is a promising substitute for a chondrocytic experimental model, displaying sufficient proliferative activity and presenting a consistent response to the phenotype of primary human chondrocytes [10]. It is often used as a model for osteoarthritis studies [11].

*Piper kadsura* (Choisy) Ohwi (Acta Phytotax. Geobot. 3(2):81 (1934)) is a perennial vine herb that grows in the forests of Taiwan and is widely used for the treatment of asthma and arthritic conditions [12]. Previous studies have shown that extracts of *P. kadsura* have anti-neuritis [13], anti-platelet activating factor [14], anti-inflammation [15], and antioxidation [16] effects, as well as other effects.

Microwave-assisted extraction (MAE) has achieved remarkable results in the areas of chemical synthesis, biomedicine, and new-material manufacturing. This technique can convert electromagnetic energy into heat at a specific frequency and temperature to rapidly heat the reaction materials. MAE can achieve high yields, selectivity, and reaction rates, and this method can use less solvent, consume less energy, and produce less waste than do traditional heating syntheses [17].

SW1353 cells showing oxidative stress induced by H_2_O_2_ possess properties similar to those of the articular chondrocytes of OA [6]. The aim of this study is to investigate the protective effects of *P. kadsura* extract (PKE) for oxidative stress to cells on OA-related symptoms and cellular apoptosis.

## 2. Results

### 2.1. The Optimal Extraction Parameters for P. kadsura by MAE

On the basis of 4 g of ground sample, the parameters in the MAE system were changed to obtain the best extraction results. Figure 1 indicates that the maximal yields of extraction were at 800 W power, 30 min extraction time, 42 °C operative temperature, and 1:15 solid/ethanol (g/mL) ratio, with the extracted yield being 11.9 ± 0.76%.

### 2.2. Purification of Crude Extract and Biological Characteristics

The crude extract of *P. kadsura* from MAE was continuously separated by liquid-liquid extraction between the polarity differences of various solvents. The order of extraction yields of different organic solvents is ethyl ether fraction > aqueous fraction > hexane fraction > ethyl acetate fraction, as shown in Table 1. Thus, most of the extracts dissolved into the ethyl ether fraction indicated that the extracts are low- to medium-polarity compounds.

Based on the calibration curve (y = 0.0087x + 0.1033, R^2^ = 0.997) of gallic acid (50–250 mg/L), the content of phenolic in crude and various solvent extracts ranged from 1256.6 ± 0.7 to 29.6 ± 0.4 mg gallic acid/g dried extract, as seen in Figure 2a. Ethyl ether extract had the greatest phenolic content, while the smallest phenolic content was found in aqueous extract. Similarly, the calibration curve (y = 0.0072x + 0.0234, R^2^ = 0.9966) of quercetin (50–250 mg/L) was employed for evaluation of the flavonoid content of crude and various solvent extracts. The maximum amount of flavonoid was found in the ethyl ether extract (10.44 ± 0.09 mg quercetin/g dried extract), while the lowest amount (1.12 ± 0.1 mg quercetin/g dried extract) was found in the aqueous extract, as shown in Figure 2b. Table 2 shows that the crude and various solvent extracts of *P. kadsura* have an obvious inhibition zone effect on most strains. Except for *P. aeruginosa* ATCC 27853 and ATCC 29260, the average inhibition zone diameter is more than 20.0 mm. Extracted by liquid-liquid distribution, the ethyl ether extract has a significant antimicrobial effect on *S. aureus* ATCC 6538P, *A. baumannii* ATCC 19606, and *E. coli* ATCC 25257 strains, while the aqueous and ethyl acetate extracts have no significant antimicrobial ability.

The concentration for SW1535 cells viability treated via various liquid-liquid extracts and H_2_O_2_ was confirmed beforehand (data not shown), and then various liquid-liquid extracts and H_2_O_2_ co-treatments were used to measure the cell viability. Figure 3a indicates that the ethyl ether extract and ethyl acetate extract resulted in higher survival rate and better cytoprotective ability than other extracts. Figure 3b shows the scavenging ability versus the different concentrations of various liquid-liquid extracts upon doing a DPPH assay. The ethyl ether extract showed the best antioxidant ability, and its EC_50_ (concentration for 50% of maximal effect) was of 7.19 ± 0.8 μg/mL, only slightly inferior to that of vitamin C (EC_50_ = 4.60 ± 0.66 μg/mL). The lower the EC_50_, the stronger the ability to capture free radicals. Therefore, the ethyl ether fraction was further purified through column chromatography based on extract yield, total polyphenols and flavonoid contents, antimicrobial activity, cell viability, and antioxidant capacity.

To choose the 6:1 hexane/ethyl acetate ratio to pure methanol as eluent liquid based on thin-layer chromatography, we separated 13 fractions from open-column chromatography. The fraction yield from open-column chromatography is shown in Table 3. The fractions with H_2_O_2_ were used to co-treat SW1353 cells for the cell viability assay. Figure 3c shows that the best cell survival was with Fraction 5. The DPPH assay of various column fractions is shown in Figure 3d, in which the best antioxidant ability was from Fraction 9 (EC_50_ = 1.77 ± 0.08 μg/mL), followed by Fraction 5 (EC_50_ = 7.9 ± 0.14 μg/mL) and Fraction 1 (EC_50_ = 7.96 ± 0.09 μg/mL). On the basis of various fraction yields, cell viability, and antioxidant capacity, Fraction 5 (as PKE) was used as the test drug in subsequent experiments. The cell viability of the PKE is dose-dependent, as shown in Figure 3e. Based on the result, the PKE is not toxic for SW1353 cells.

### 2.3. Apoptosis Assay and Cell Morphology Change

Cellular apoptosis in SW1353 cells treated with PKE (60 μg/mL), PKE (60 μg/mL) with 3.6 mM added H_2_O_2_, and 3.6 mM H_2_O_2_ for 24 h was determined according to cellular vitality. Cellular morphology was observed in SW1353 cells treated with PKE or H_2_O_2_ for 24 h, as shown in Figure 4a. The cells treated with PKE showed normal morphology, but the cells treated with H_2_O_2_ shrank significantly, and apoptotic bodies were observed. However, cell shrinkage or apoptotic bodies were not obvious for cells treated with PKE with added H_2_O_2_. Hence, PKE has significant ability to resist oxidative stress. The proportions of apoptotic SW1353 cells were 10.5%, 18.0%, and 28.4% under treatment with PKE, PKE with added H_2_O_2_, and H_2_O_2_, respectively, and that for the control was 10.0%, as seen in Figure 4b. This result indicates that H_2_O_2_ can cause cell apoptosis, but PKE protects cells from oxidative stress. The viability of L929 fibroblast cells treated with 60 μg/mL of PKE for 24 h and 48 h were 94.67% and 108.27%, respectively. Thus, PKE is not toxic to normal non-cancerous cells.

### 2.4. ROS Assay

SW1353 cells were treated with PKE (60 μg/mL), PKE (60 μg/mL) with 3.6 mM added H_2_O_2_, and 3.6 mM H_2_O_2_ for 24 h, and then cells were stained with CM-H2DCFDA to measure fluorescent expression. Table 4 indicates that the maximal expression of ROS (73.73 ± 1.54%) was in cells treated with H_2_O_2_, but PKE protects SW1353 cells from ROS and reduces ROS intensity to 70.94 ± 0.92%.

### 2.5. qRT-PCR

The gene expression of the antioxidant enzyme SOD-2 can be upregulated by H_2_O_2_ and PKE simultaneously (Figure 5a). The upregulation expression of PKE is 1.2-fold higher than that of H_2_O_2_ only. Both the gene expression of GPx and CAT were downregulated by the H_2_O_2_ treatment, but the expression by PKE treatment was upregulated 1.26- and 3.69-fold in Figure 5b,c, respectively. The Bcl-2/Bax ratio was significantly upregulated by treatment with PKE and was downregulated slowly by treatment with H_2_O_2_; however, the expression of Bcl-2/Bax was still increased by PKE combined with H_2_O_2_ treatment simultaneously, as seen in Figure 5d. Figure 5e indicates that caspase-3 expression was increased by the H_2_O_2_ treatment, but the expression was downregulated by PKE treatment. These results reveal that H_2_O_2_-treated SW1353 cells show induced programmed cellular death via the mitochondrial pathway, and that PKE can slow the apoptosis by regulating the antioxidant enzymes SOD-2, CAT, and GPx to protect cells.

Since H_2_O_2_ is regulated by the MAPK family, inhibitors were added to clarify pathways that protect cells by PKE treatment. Figure 6 indicates that Bax expression was downregulated significantly by addition with the inhibitors, such as SB 203580-p38 MAPK, U0126-MKK, and SP600125-JNK, and the expressions of Bcl-2 and SOD-2 were upregulated.

### 2.6. Western Blot

Western blot assay was used to measure the changes in the PARP concentrations of SW1353 cells after PKE or H_2_O_2_ treatment. PARP cleavage was induced by H_2_O_2_, as seen in Figure 7. PKE can prevent the breakage of PARP caused by H_2_O_2_. The fracture in the H_2_O_2_ group is 1.81-fold that of the group with PKE added and H_2_O_2_ co-treatment. These results indicate that PKE-induced apoptosis could block the caspase-dependent apoptosis pathway, which involves the effector caspase-3.

### 2.7. Composition Identification

Figure 8a shows that the PKE extract contains multiple peak signals. After manually comparing mass diagrams, we speculate on the structure of the substance and its mass result. Figure 8b is the structure of bicyclo [2.2.1] heptan-2-ol, 1,7,7-trimethyl-, (1S-endo)- (C_10_H_18_O), Figure 8c is alpha-humulene (C_15_H_24_), Figure 8d is hydroxychavicol (C_9_H_10_O_2_), and Figure 8e is the dry *Piper kadsura*.

## 3. Materials and Methods

### 3.1. Preparation of the Extracts of P. kadsura

*P. kadsura* was purchased from a local grass shop in Taiwan, and then all the samples were crushed into powders of less than 0.30 mm. Different parameters (operating temperature, extraction time, solid/liquid (g/mL) ratio, and microwave power) were adjusted to optimize the process conditions for maximum yield of extraction. Four-gram powders of *P. kadsura* were mixed with 95% ethanol and extracted using a microwave-assisted machine (MAS-II Plus, Sineo, Shanghai, China) to determine the best yield of the ethanol extract. The ethanol extract was extracted successively by liquid-liquid extraction with hexane, ethyl ether, and ethyl acetate to obtain a hexane extract, ethyl acetate extract, ethyl acetate extract, and aqueous extract, respectively. On the basis of results of cell viability and antioxidant activity from various liquid-liquid extracts, the ethyl ether extract has the best cell survival rate and antioxidant capacity. Therefore, the ethyl ether extract of *P. kadsura* was subjected to column chromatography with silica gel (230–400 mesh) using hexane-ethyl acetate (6:1), hexane-ethyl acetate (3:1), hexane-ethyl acetate (1:1), and ethyl acetate, and methanol extracting solvent was then used as mobile phase to collect 13 pooled fractions based on their TLC profiles. These various fractions were tested for cell viability and antioxidant activity.

### 3.2. Biological Characteristics of P. kadsura Extract

#### 3.2.1. Total Polyphenols and Flavonoid Contents Determination

The total polyphenols of crude and various liquid-liquid extracts of *P. kadsura* were determined according to the method of Folin–Ciocalteu [18] with some modification. A total of 200 µL of extract (1000 mg/L) or gallic acid (50–250 mg/L) was mixed with 1.0 mL of Folin–Ciocalteu reagent (0.5 M) and 1.0 mL of sodium carbonate solution (75 g/L), after 2 h standing in the dark at room temperature, the optical density was measured at 760 nm against a blank. The total phenolic contents were calculated on the basis of the calibration curve of gallic acid and expressed as gallic acid equivalents, in mg gallic acid per gram of the dried extract.

The flavonoid contents of crude and various liquid-liquid extracts were measured following the Dowd method [19]. An aliquot of 1 mL of extract solution (1000 mg/L) or quercetin (50–250 mg/L) were mixed with 0.1 mL of 10% (*w*/*v*) AlCl_3_ solution in methanol, 0.1 mL potassium acetate (1.0 M), and 5.6 mL distilled water. The mixture was incubated for 45 min at room temperature, followed by the measurement of absorbance at 415 nm against the blank. The total flavonoid contents were calculated on the basis of the calibration curve of quercetin and expressed as quercetin equivalents, in mg quercetin per gram of the dried extract.

#### 3.2.2. Antimicrobial Activity Assay—Disc Diffusion Method

The five standard strains (A. baumannii ATCC 19606, S. aureus ATCC 6538P, P. aeruginosa ATCC 27853, P. aeruginosa ATCC 29260, E. coli ATCC 25257) were purchased from the Bioresource Collection and Research Center (Hsinchu, Taiwan) and inoculated into 5.0 mL LB broth and cultivated in a 200 rpm 37 °C incubator for 12 to 16 h. A total of 50.0 μL strains solution (3 × 10^8^ CFU/mL) was added to 5.0 mL LA soft agar, vortexed, and mixed evenly, then poured on the LA medium to form a double layer. We took 30.0 μL of crude and various extracts solution (0.1 g/mL) into a sterile 6 mm filter paper disk, with DMSO (dimethyl sulfoxide) as the negative control group and tetracycline (7.50 mg/mL) as the positive control group. All plates were incubated at 37 °C for 12–16 h, observed, and measured for the size of the disk inhibition zone (DIZ).

#### 3.2.3. Antioxidant Activity Analysis—DPPH Free Radical Scavenging Activity

The 2-2-diphenyl-β-picrylhydrazyl (DPPH) free radical scavenging test was performed according to the method of Huang et al. [20] with some modifications. Samples (100 μL each) of different concentrations were added to 25 μL of 0.5 mM DPPH solution in ethanol, mixed uniformly, and then allowed to stand in the dark for 30 min, followed by spectroscopy at 517 nm. Double-distilled water served as the negative control, and vitamin C served as the positive control. The radical-scavenging effect was calculated using the following equation:Scavenging effect (%) = [1 − (A_sample_ − A_blank_)/A_control_] × 100%(1)
where A_sample_ is the sample group absorbance, A_blank_ is the blank absorbance, and A_control_ is the absorbance of double-distilled water.

### 3.3. Antioxidative Activity In Vitro

Using SW1353 cells as test subjects, we investigated the protective effects of extracts from *P. kadsura* on its cytotoxicity and oxidative damage. SW1353 cell was purchased from the Bioresource Collection and Research Center (Hsinchu, Taiwan). These cells were cultured as monolayers in Leibovitz’s L15 medium containing 10% FBS, 1% penicillin, and streptomycin, and placed in a humidified incubator under 0% CO_2_ at 37 °C for ordinary cultures. The culture medium was changed every 2–3 days.

#### 3.3.1. Cell Survival Assay and Cell Morphology

The number of surviving cells after treatment with an extract was measured in terms of the metabolic activity of mitochondrial enzymes, which was based on 3-(4,5-dimethylthiazol-2-yl)-2,5-diphenyltetrazolium bromide (MTT). SW1353 cells and L929 fibroblasts were treated with different concentrations of various extracts and hydrogen peroxide for 24 h, and then 5 mg/mL MTT was added at 37 °C for 3 h. After the medium was removed, the purple needle-like crystals were dissolved in DMSO, and the absorbance was measured at wavelength of 570 nm using a spectrophotometer (Multiskan FC, Thermo Fisher Scientific, Waltham, MA, USA). From the results of cell viability from various liquid-liquid extracts and hydrogen peroxide, we chose the PKE as the Fraction 5 of the ethyl ether extract from column chromatography and hydrogen peroxide with co-treatment for further study.

SW1353 cells were cultured initially on six-well tissue culture plates, treated with the PKE (60 μg/mL), PKE (60 μg/mL) with 3.6 mM added H_2_O_2_, and 3.6 mM H_2_O_2_ for 24 h (based on the optimal results of cell viability). The cells were fixed in situ in 4% paraformaldehyde for 10 min to facilitate observation of the cellular morphology. All specimens were examined using a light microscope (CKX53, Olympus).

#### 3.3.2. Apoptosis Assay

SW1353 cells were seeded into six-well tissue culture plates (2 × 10^5^ per well), and then PKE (60 μg/mL), PKE (60 μg/mL) with 3.6 mM added H_2_O_2_, and 3.6 mM H_2_O_2_ were treated for 24 h. The cells were removed from the culture plates and centrifuged at 447.2× *g* for 5 min. The supernatant was removed, and the precipitate was suspended in 200 μL of 1 × PBS. The suspension was mixed continuously while 800 μL of cold ethanol was added, and the cells were stored at −20 °C overnight. The extent to which the levels of thiols had decreased indicated overall cell health following extract treatment, according to the manufacturer’s protocol for the cell vitality kit (ChemoMetec A/S, Allerod, Denmark). Cell apoptosis assays were detected using a NucleoCounter NC-250 (ChemoMetec A/S, Allerod, Denmark), and these results can explain the cellular apoptosis.

#### 3.3.3. ROS Assay

SW1353 cells were seeded into six-well tissue culture plates (2 × 10^5^ per well), and then PKE (60 μg/mL), PKE (60 μg/mL) with 3.6 mM added H_2_O_2_, and 3.6 mM H_2_O_2_ were treated for 24 h. The cells were removed from the culture plates and centrifuged at 113.4× *g* for 5 min. The supernatant was removed, and the precipitate was suspended in 200 μL of 1×PBS. The suspension was mixed continuously while 400 μM of CM-H2DCFDA was added (control group was replaced by 1×PBS) to allow reaction for 30 min in the dark. Post-reaction cells were centrifuged at 188.9× *g* for 5 min, and then the precipitate was suspended in 1000 μL of PBS for analysis by flow cytometry (BD FACSCalibur^TM^ Dual Laser, Becton, Dickinson, Franklin Lakes, NJ, USA) at 480 nm for the excited state and 530 nm for the emission state.

#### 3.3.4. Quantitative Real-Time PCR

SW1353 cells (2 × 10^5^ per well) were seeded into six-well tissue culture plates, and then PKE (60 μg/mL), PKE (60 μg/mL) with 3.6 mM added H_2_O_2_, and 3.6 mM H_2_O_2_ were treated for 24 h. After the medium was removed, the total RNA of cells was extracted using TRIzol reagent (Ambion^®^, Life Technologies™, USA) according to the manufacturer’s operating manual. RNA quantity was assessed with Nanodrop One/One^C^ Spectrophotometer (Thermo Scientific instruments). cDNA was synthesized by reverse transcription using Super Script^®^ III Reverse Transcriptase kit (Life Technologies™, USA) following protocols from Invitrogen™ Corporation (USA). Real-time polymerase chain reaction (RT-PCR) was performed using Smart Quant Green Master Mix according to protocols from Protech Technology Enterprise Co., Ltd. (Taiwan). An initial denaturation was performed at 65 °C for 10 min followed by 60 cycles of 55 °C for 30 min and 85 °C for 5 min. The relative gene-expression fold change was determined through the 2^−^^△△Ct^ method and normalized to transcripts of the housekeeper gene GAPDH. The used PCR primers were purchased from Genomics (Taiwan) and listed below: Bcl-2 (Forward: 5′-GAG ACA GCC AGG AGA AAT CA-3′, Reverse: 5′-CCT GTG GAT GAC TGA GTA CC-3′); Bax (Forward: 5′-CAT CTT CTT CCA GAT GGT GA-3′, Reverse: 5′-GTT TCA TCC AGG ATC GAG CAG-3′); Caspase3 (Forward: 5′-CTC GGT CTG GTA CAG ATG TCG A-3′, Reverse: 5′-CAT GGC TCA GAA GCA CAC AAA C-3′); CAT (Forward: 5′-AGA GGA AAC GCC TGT GTG AG-3′, Reverse: 5′-TAG TCA GGG TGG ACG TCA GT-3′); SOD-2 (Forward: 5′-GTG TCT GTG GGA GTC CAA G-3′, Reverse: 5′-TGC TCC CAC ACA TCA ATC CC-3′); GPx (Forward: 5′-GGA CTA CAC CCA GAT GAA-3′, Reverse: 5′-GTG GCG TCG TCA CTT G-3′); GAPDH (Forward: 5′-ATG AGA AGT ATG ACA ACA GCC-3′, Reverse: 5′-AGT CCT TCC ACG ATA CCA AA-3′).

To investigate the effect of hydrogen peroxide on oxidative stress in SW1353 cells, 100 µL of 10 µM inhibitors of different MAPKs families (SB 203580, U0126, and SP600125) was added to the medium for 2 h in culture. Subsequently, these cells were treated with PKE (60 μg/mL), PKE (60 μg/mL) with 3.6 mM added H_2_O_2_, and 3.6 mM H_2_O_2_ for 24 h, and then the gene expression of Bax, Bcl-2, and SOD-2 was analyzed by qRT-PCR.

#### 3.3.5. Western Blotting

SW1353 cells (1.2 × 10^6^ per dish) were cultured overnight, and then added PKE (60 μg/mL), PKE (60 μg/mL) with 3.6 mM added H_2_O_2_, and 3.6 mM H_2_O_2_ were treated for 24 h. After the medium was removed, the total cell protein was extracted by lysing the cells in a buffer containing 50 mM Tris-Cl, 150 mM NaCl, 0.1% Triton X-100, 0.1% (*m*/*v*) SDS, 1.0 mM sodium orthovanadate, and 1 mM NaF at pH 8.0. Protein concentrations were determined by Bradford protein assay at an absorbance of 595 nm. Equal amounts of proteins were subjected to 10% SDS-polyacrylamide gel (SDS-PAGE), electrically transferred onto polyvinylidene fluoride (PVDF) membranes (Millipore, Billerica, MA, USA), blocked with blocking buffer (5% skimmed milk powder in PBS), and then probed with primary antibodies (PARP; 1:1000; Cell Signaling Technology, Danvers, MA, USA) overnight at 4 °C. Blots were washed and incubated for 1 h with IRDye800^®^-conjugated secondary antibodies (1:8000; Cell Signaling Technology, USA), and then washed with 0.1% Tween-20 in Tris-buffered saline (*TBS*). The proteins were visualized via chemiluminescence using the Amersham ECL Plus Western Blotting Detection kit (GE Healthcare, Chicago, IL, USA) according to the manufacturer’s instructions. β-Actin (Cell Signaling Technology, USA) was used as the loading control for protein expression in the treated cells.

### 3.4. Composition Identification

The exact components of PKE were identified by gas chromatography–mass spectrometry (JEOL, AccuTOF GCX). PKE (1.25 mg dissolved in 0.5 mL methanol) was taken, and then 0.5 µL was injected into the column (Rxi-5MS, 30 m, I.D. 0.25 mm, 0.25 µm film). Initial temperature was 40 °C; it was heated to 300 °C at a rate of 10 °C per minute and retained for 10 min. The carrier gas of the column was helium, and the flow rate was 1 mL/min.

### 3.5. Statistical Analysis

All experiments with multiple samples were performed thrice. Data were presented as mean ± standard deviation (SD), and ANOVA was used to treat data. Statistical comparisons were performed, and comparisons with p values smaller than 0.05 were considered significant.

## 4. Discussion

Ethanol and water are often used for traditional Chinese herbal medicine extraction. However, the preservation of water extracts is more difficult than that of ethanol extracts. Water also easily extracts more sugars, starches, and other substances, which makes the boiling point of the extract increase or gelatinize the extract, resulting in difficulties in subsequent concentration and drying. Ethanol has a higher selectivity than water and can effectively obtain plant secondary metabolites [21]; hence, 95% ethanol was used as extraction solvent in this study.

The extracted substance decreases with the increase of the extraction time after 30 min by MAE. The reason for this may be the degradation of volatile oils in plants [22]. In addition, the extraction yields begin to decrease when the temperature is higher than 42 °C and the power exceeds 800 W. It may be because the viscosity and surface tension of the solvent can be reduced at high temperatures to promote accelerated extraction, but it could also be the degradation of volatile oils in plants [23,24]. Natural plant compounds are highly compatible with organic solvents because of their structure and polarity, so their bonding ability is also different with various solvents [21]. Therefore, different types of compounds can be assigned to different solvents to achieve the effect of separation by washing with different eluents from low- to high-polarity solvent. In this study, from the lowest polarity (n-hexane) to the most highly polar solvent (ethyl acetate), the different substances were separated into the solvents, and the substances that could not be distributed to the organic solvent were left in the aqueous phase. Most of the polyphenols (including flavonoids) exist in the ethyl ether extraction, which also has the greatest antimicrobial ability. It was found that most active substances were first dissolved in the ethyl ether solvent and then mixed with different polar solvents for column chromatography and TLC to separate them into 13 fractions. These fractions were tested for cell viability and antioxidant activity, and the most effective components in Fraction 5 were confirmed to be PKE.

Oxidative damage can lead to degenerative diseases and chronic non-infectious diseases. However, antioxidants from natural plants have showed significant cellular protection properties. Pepper plants can be found in tropical and subtropical regions of the world. About 2000 different types of pepper plants have been counted, and piper is the largest species. Most of them have natural antioxidant activity [25,26,27,28]. Here, we observed that the crude extract of *P. kadsura* has good antioxidant activity, as indicated by the DPPH test, and that the antioxidant capacity of the extract can be promoted through the purification process. As a result, it was found that the antioxidant activity of various fractions separated from the ethyl ether extract after column chromatography purification were more than 50%, with EC_50_ values less than 20 μg/mL; three of them were below 10 μg/mL. Through the increase of ROS, H_2_O_2_ can be used for cellular ROS induction that provides free radicals to simulate the situation when the load capacity of cells is exceeded in the organism [29]. The increase of ROS in chondrocytes is considered to be related to the dysfunction of cells and the degradation of articular cartilage, which may lead to cartilage aging or arthritis. When cells encounter harmful conditions caused by ROS, the cells induce antioxidant enzymes such as SOD, CAT, and GPx as a compensation mechanism to ensure the integrity of the cells [30]. SW1353 cells treated with H_2_O_2_ alone could regulate antioxidant enzyme SOD-2, CAT, and GPx to promote cell apoptosis and stimulate downstream-of-MAPK-family protein kinases. This then regulates Bax and Bcl-2 on the mitochondrial membrane and then activates caspase-3, and cuts off PARP, leading to cell death. Activated MAPKs participate in various signal stimulations in induced chondrocyte apoptosis, which mediate cell survival and proliferation; the results after regulation are related to cartilage degenerative diseases. The combined treatment of PKE and H_2_O_2_ can reverse the effect of H_2_O_2_ alone and effectively slow down the consumption of antioxidant enzymes CAT, GPx, and SOD-2 by H_2_O_2_, thus increasing Bcl-2/Bax expression and regulating the performance of caspase-3 and PARP. Through the addition of inhibitors, it is found that all of the MAPKs are the key to regulation. The three pathways of JNK [1], MEK/ERK [31], and p38 [32] are regulated by regulating GPx, CAT, and SOD-2 in cells to protect the regulation of Bcl-2 and Bax on the mitochondrial membrane. PKE was found to promote cell proliferation in the MTT test, and it was confirmed in the results of the cell morphology observation and cell apoptosis analysis. The performance of SOD-2, CAT, and GPx genes was enhanced in the qRT-PCR assay. H_2_O_2_ itself is related to various inflammatory diseases of the body, which have the ability to induce cell inflammation [6,33,34]. In this study, the relationship between cells and MAPK was used to clarify the possible pathways of PKE, as shown in Figure 9. It has not been thoroughly explored whether PKE has full anti-inflammatory ability. It is only confirmed through the gene end of the cell that MAPKs and antioxidant enzymes CAT are indeed regulated by H_2_O_2_, and that PKE can indeed alleviate the damage caused by H_2_O_2_ and protect the cells against inflammation response.

The main anti-inflammatory composition of the PKE was analyzed by GC-MS, and it was determined that there were bicyclo [2.2.1] heptan-2-ol-1,7,7-trimethyl-, (1S-endo)-, C_10_H_18_O, alpha-humulene, C_15_H_24_ and hydroxychavicol, and C_9_H_10_O_2_. Bicyclo [2.2.1] heptan-2-ol-1,7,7-trimethyl-, (1S-endo)- is a stereoisomer of borneol, which has been proven to have cytoprotective effects in the literature [35,36]. However, if the concentration is greater than 2 mM, it may cause damage to cells. Alpha-humulene is a sesquiterpene found in many plants that has been shown to protect nerve cells [37] and have anti-inflammatory effects [38]. Hydroxychavicol is a polyphenol and has good antioxidant capacity [39]. It is also a xanthine oxidase inhibitor and can be used clinically to treat hyperuricemia [40].

## 5. Conclusions

Biologically active substances contained in *P. kadsura* were obtained by extraction and column chromatography. The optimum conditions for MAE were 42 °C, 800 W, 30 min, 1 g piper, and 15 mL of 95% ethanol. PKE had good DPPH removal efficiency, and it could effectively protect cells from H_2_O_2_ attack and prevent cell death. The substances of PKE that can protect the cells may be bicyclo [2.2.1] heptan-2-ol-1,7,7-trimethyl-(1S-endo)-, alpha-humulene, and hydroxychavicol, which were shown to be present via GC–MS identification. SW1353 chondrosarcoma cells were significantly protected from H_2_O_2_ assault by PKE, likely via regulation of MAPKs signals and modulation of mitochondrial pathways.

## Figures and Tables

**Figure 1 molecules-26-06287-f001:**
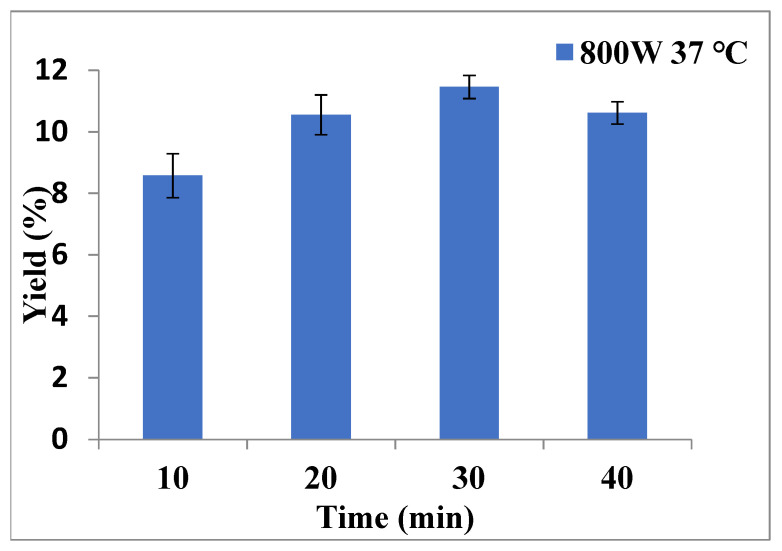
The yields of various parameters in the MAE; the maximal yield of extracts were at 800 W power, 30 min extract time, 42 °C operative temperature, and 1:15 solid/ethanol (g/mL) ratio.

**Figure 2 molecules-26-06287-f002:**
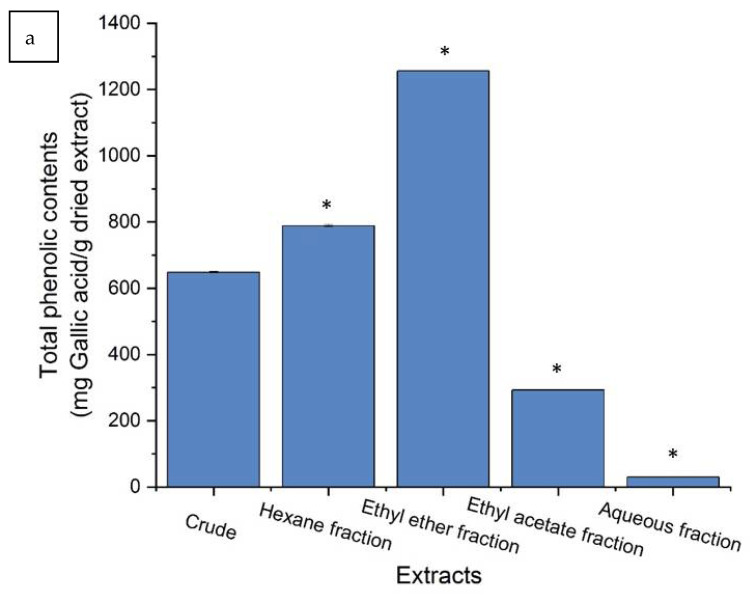
The total phenolic acids (**a**) and flavonoid contents (**b**) in crude and various solvent extracts of *P. kadsura*. Values are expressed as mean ± SD (*n* = 3), * *p* < 0.05 for compared with crude group.

**Figure 3 molecules-26-06287-f003:**
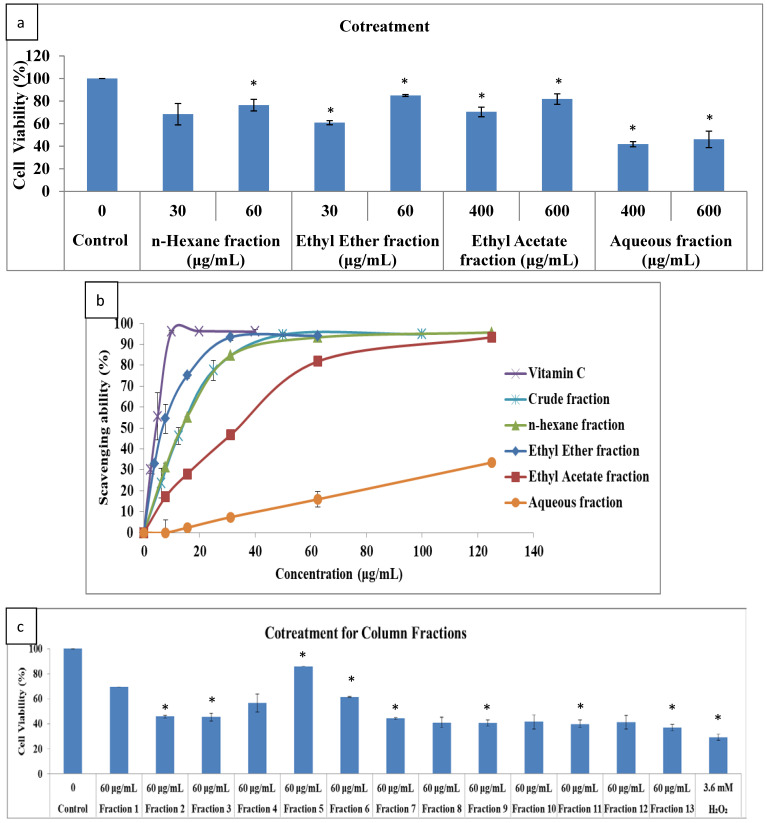
The cell viability for various liquid-liquid extracts and H_2_O_2_ co-treatment with different concentrations (**a**), the DPPH assay treated with various liquid-liquid extracts (**b**), cell viability for various fraction extracts and H_2_O_2_ co-treatment (**c**), the DPPH assay treated with extracts of various fractions (**d**), and the dose-dependent cell viability of the PKE (**e**). Values are expressed as mean ± SD (*n* = 3), * *p* < 0.05 for compared with control group or 0 concentration.

**Figure 4 molecules-26-06287-f004:**
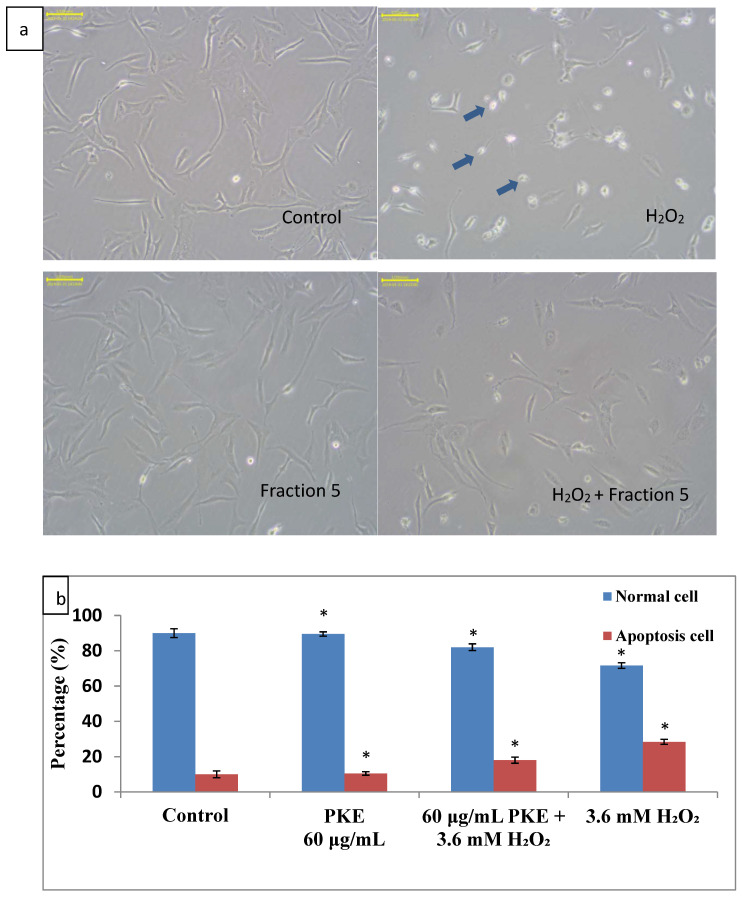
Cellular morphology of SW1353 treated with PKE or H_2_O_2_ for 24 h (the arrows are the apoptotic bodies) (**a**); the proportion of apoptotic SW1353 cells treated with PKE extract (60 μg/mL), PKE (60 μg/mL) with 3.6 mM added H_2_O_2_, and 3.6 mM H_2_O_2_ for 24 h (**b**). scale bar = 100 μm. Values are expressed as mean ±SD (*n* = 3), * *p* < 0.05 for compared with control group.

**Figure 5 molecules-26-06287-f005:**
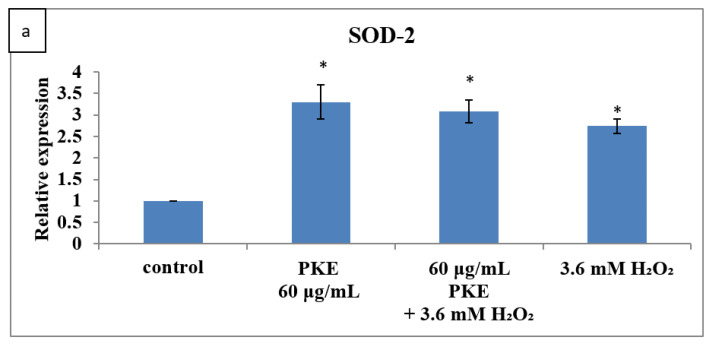
The gene expression of the antioxidant enzymes SOD-2 (**a**), GPx (**b**), and CAT (**c**), as well as the Bcl-2/Bax ratio (**d**), and caspase-3 (**e**) after cells were treated with PKE extract (60 μg/mL), PKE (60 μg/mL) with 3.6 mM added H_2_O_2_, and 3.6 mM H_2_O_2_ for 24 h. Values are expressed as mean ±SD (*n* = 3), * *p* < 0.05 for compared with control group.

**Figure 6 molecules-26-06287-f006:**
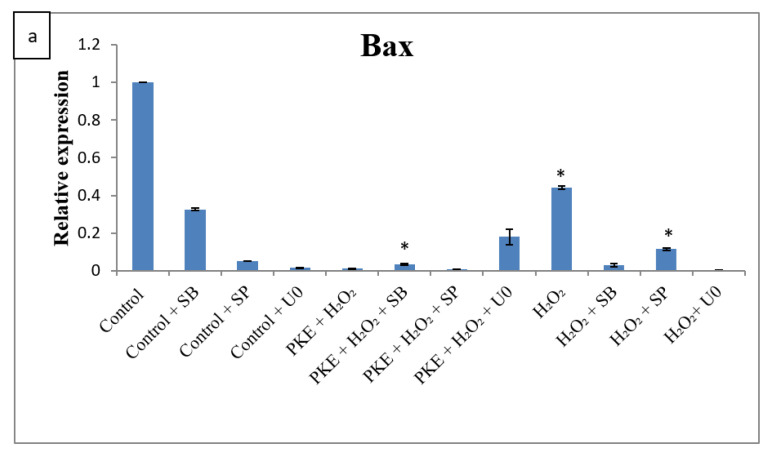
The gene expression of Bax (**a**), Bcl-2 (**b**), SOD-2 (**c**), and Bcl-2/Bax (**d**) after SW1353 was treated with MAPK inhibitors with added PKE or H_2_O_2_. Values are expressed as mean ± SD (*n* = 3), * *p* < 0.05 for compared with control group.

**Figure 7 molecules-26-06287-f007:**
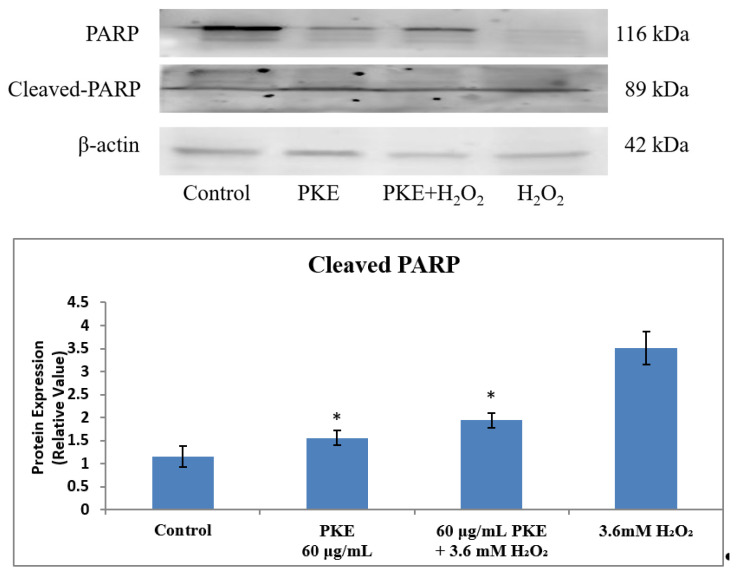
The cleaved PARP in the western blot assay after cells were treated with PKE extract (60 μg/mL), PKE (60 μg/mL) with 3.6 mM added H_2_O_2_, and 3.6 mM H_2_O_2_ for 24 h. Values are expressed as mean ± SD (*n* = 3), * *p* < 0.05 for compared with control group.

**Figure 8 molecules-26-06287-f008:**
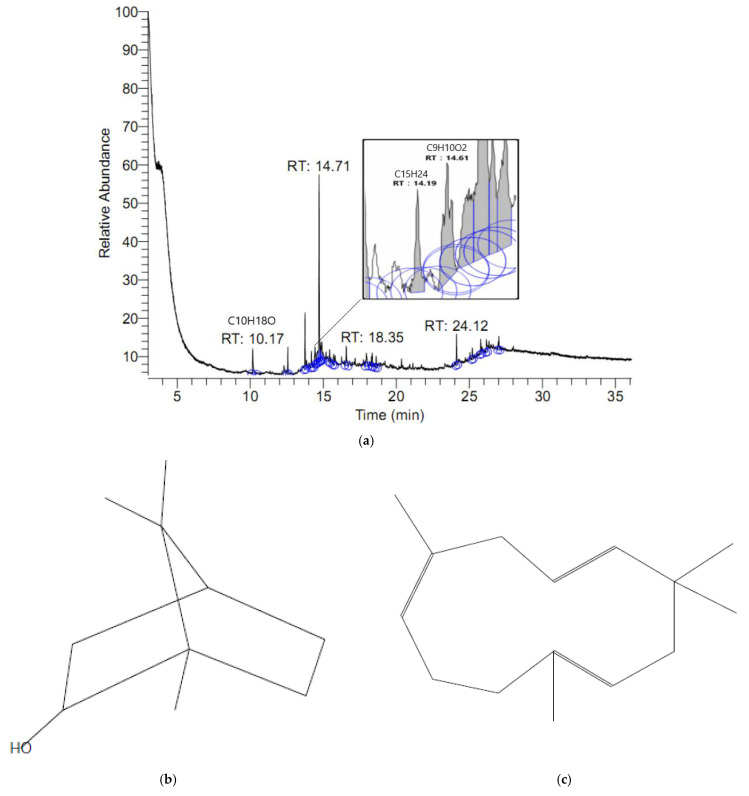
The peak signals of PKE from GCMS analysis (**a**), the structure of bicyclo[2.2.1]heptan-2-ol, 1,7,7-trimethyl-, (1S-endo)- (**b**), alpha-humulene (**c**), hydroxychavicol (**d**), and dry *Piper kadsura* (**e**).

**Figure 9 molecules-26-06287-f009:**
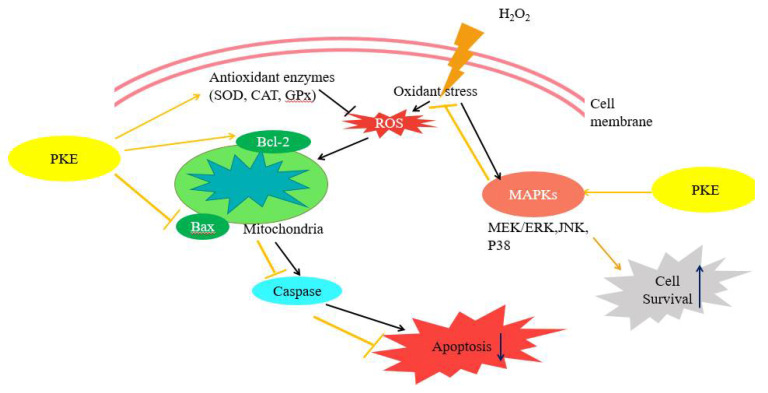
The possible routes by which PKE protects SW1353 from oxidative stress through JNK, MEK/ERK, and p38 pathways.

**Table 1 molecules-26-06287-t001:** The yield of different liquid-liquid separation.

Extract	Yield (%)
Crude fraction	11.90 ± 0.54
n-Hexane fraction	1.69 ± 0.36
Ethyl ether fraction	4.40 ± 0.20
Ethyl acetate fraction	0.44 ± 0.06
Aqueous fraction	2.73 ± 0.15

**Table 2 molecules-26-06287-t002:** Antimicrobial activity of crude and various liquid-liquid extracts of *P. kadsura*.

Strains	DIZ (mm)					
	Crude	Hexane	Diethyl Ether	Ethyl Acetate	Aqueous	Tetracycline
	1000.0 (mg/mL)	100.0 (mg/mL)	100.0 (mg/mL)	100.0 (mg/mL)	100.0 (mg/mL)	7.50 (mg/mL)
*P. aeruginosa* ATCC 27853	22.60 ± 0.90	14.28 ± 1.73	16.05 ± 3.45	10.75 ± 0.55	6.25 ± 0.25	15.50 ± 0.20
*P. aeruginosa* ATCC 29260	14.20 ± 0.20	9.20 ± 1.20	14.23 ± 0.23	7.23 ± 0.23	–	14.20 ± 0.20
*S. aureus* ATCC 6538P	25.73 ± 1.73	20.75 ± 0.75	20.75 ± 1.25	12.50 ± 0.50	7.45 ± 0.03	31.93 ± 2.43
*A. baumannii* ATCC 19606	23.50 ± 1.50	21.03 ± 1.33	26.20 ± 0.28	8.48 ± 0.05	–	18.00 ± 0.00
*E.coli* ATCC 25257	21.70 ± 0.50	19.50 ± 2.00	22.15 ± 0.15	7.38 ± 0.05	–	26.08 ± 0.53

Data = mean ± SD; –:no active.

**Table 3 molecules-26-06287-t003:** The yield of various fraction of column separation.

Extract	Elution	Yield (%)
Fraction 1	Hex:EA = 6:1	1.73 ± 0.31
Fraction 2	Hex:EA = 6:1	0.21 ± 0.01
Fraction 3	Hex:EA = 3:1	0.06 ± 0.00
Fraction 4	Hex:EA = 3:1	0.17 ± 0.01
Fraction 5	Hex:EA = 1:1	0.33 ± 0.03
Fraction 6	Hex:EA = 1:1	0.52 ± 0.09
Fraction 7	Hex:EA = 1:1	0.11 ± 0.01
Fraction 8	Hex:EA = 1:1	0.21 ± 0.01
Fraction 9	EA = 1	0.34 ± 0.03
Fraction 10	EA = 1	0.21 ± 0.01
Fraction 11	EA = 1	0.15 ± 0.00
Fraction 12	EA = 1	0.09 ± 0.00
Fraction 13	MeOH = 1	0.64 ± 0.08

n-Hexane: Hex, ethyl acetate: EA, methanol: MeOH.

**Table 4 molecules-26-06287-t004:** The ROS content of different treatments.

Treated Drug	ROS Content(%)
60 μg/mL Fraction 5	67.44 ± 1.08
60 μg/mL Fraction 5 + 3.6 mM H_2_O_2_	70.94 ± 0.92
3.6 mM H_2_O_2_	73.73 ± 1.54

## Data Availability

Not applicable.

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
