# Peer review of "Biological and Cytoprotective Effect of Piper kadsura Ohwi against Hydrogen-Peroxide-Induced Oxidative Stress in Human SW1353 Cells"

_molecules, 2021, doi:10.3390/molecules26206287_

Round 1

Reviewer 1 Report

In  this paper, Piper kadsura Ohwi is a traditional Chinese medicine that used as a treatment for rheumatic pain was investigated regarding cell assays. Antioxidant tests and lowering the apoptosis of SW1353 cells treated with H2O2  plus upregulation of gene expression of antioxidant enzymes (SOD-2, GPx, and CAT) and the Bcl-2/Bax ratio, as well as regulation of PARP, thus conferring resistance to H2O2 attack, was proposed for the green extracts plus antimicrobial tests.  The identified components of PKE were bicycle [2.2.1] heptan-2-ol-1,7,7-trimethyl-,(1S-endo)- and alpha-humulene by gas chromatography mass spectrometry.

Weakness: only GC MS is not enough to probe biological compounds, please run HPLC MS and UV detection for all compounds, and also validation and quantitation would be a plus. 2- please check english language. 3-please add pictures of piper

Author Response

Reviewer#1:

  1. only GC MS is not enough to probe biological compounds, please run HPLC MS and UV detection for all compounds, and also validation and quantitation would be a plus.

Response: Although PKE is obtained through liquid-liquid extraction and column chromatography, it is still a mixture of multiple components. This study is only to find out the components that may have a protective effect on free radical-induced chondrocyte damage, not the isolation and identification of each component of PKE. In the future research, if the identification of the components will refer to the recommendations of the reviewer, analyze with HPLC MS and UV, and use NMR to confirm that the molecule formula and chemical structure.

  1. please check english language.

Response: We have carefully checked the engilish language of the manuscript.

  1. please add pictures of piper.

Response: We have added the picture of piper in Fig.8d.

Reviewer 2 Report

In this paper the authors, Te-Yang Huang , Chih-Chuan Wu , Wen-Ta Su, show studies about the anti inflammatory, antioxidant and antimicrobial effects of exctracts of Piper Kadsura, a plant commercially available in local grass shop in Taiwan. They have studied the characteristics of ethanol crude extracts of this plant and also those after extraction in different organic solvents with water. After characterization of best extract fraction, that results in ethyl ether thay have also purified the crude from this solvent ant tried to solve the structure of the fraction with the more antioxidant activity.

The paper reported a lot of good results about the antioxidant, antiinflammatory and antimicrobial activity of the extracts of Piper Kadsura, maybe it is poor about the resolution of structure of molecule with the best activity. A H-NMR and C-NMR could help to best characterize the molecule, confirming which reported in the text. However, the reported results are exhaustive and the paper can be published after a deep search and correction of errors and text editing such as: 

"Table 2 shows that the crude and various solvent extracts of P. kad- 237
sura has obvious inhibition zone effect on most strain" at line 238

"It can be that the reason may be the degradation of volatile oils in plants" at line 375

Author Response

Reviewer#2:

  1. "Table 2 shows that the crude and various solvent extracts of P. kadsura has obvious inhibition zone effect on most strain" at line 238

Response: We have revised the statement, as reviewer’s suggestion.

  1. "It can be that the reason may be the degradation of volatile oils in plants" at line 375

Response: We have revised the statement, as reviewer’s suggestion.

Round 2

Reviewer 1 Report

Strength Oxidative stress plays a role in regulating a variety of physiological functions in living organisms and in the pathogenesis of articular cartilage diseases. Piper kadsura Ohwi traditional Chinese medicine used as a treatment for rheumatic pain green extracts were tested regarding cell protection. And antiox activity  the extracts decreased the apoptosis of SW1353 cells treated with H2O2 and could upregulate the gene expression of antioxidant enzymes (SOD-2, GPx, and CAT) and the Bcl-2/Bax ratio, as well as regulate PARP, thus conferring resistance to H2O2 attack. The identified components of PKE were bicycle [2.2.1] heptan-2-ol-1,7,7-trimethyl-,(1S-endo)- and alpha-humulene by gas chromatography mass spectrometry.

Weakness: antioxidant activity is due to phenolic compounds I cannot see any phenolic compound detected

Please use hplc ms to detect and quantify main phenolics in the plant with ROS antioxidant activity.please check English language and journal formatting

Author Response

Several components of PKE have been identified by GC-MS, except Bicyclo[2.2.1]heptan-2-ol,1,7,7-trimethyl-(1S-endo)- and alpha-Humulene, which have been previously confirmed to have cytoprotective effect. Hydroxychavicol is also present in PKE. It belongs to polyphenols and has good antioxidant capacity. It has been written into the article and related references have been added.

Round 3

Reviewer 1 Report

only GC MS is not enough since little compounds have been detected and informed, please do a little more work to improve quality of paper,  to probe biological compounds, please run HPLC MS and UV detection for all compounds, and also validation and quantitation would be a plus.